# DNA-Net: Genetic-Inspired Dual-Chain Learning for Medical Image Domain Generalization without Negative Transfer

## Abstract

Domain Generalization (DG) in medical image segmentation remains a challenging yet essential problem, particularly when aiming to avoid negative transfer between source and target domains. The prevalent domain shift in clinical datasets often limits deep learning models' generalization beyond the source domain, and many existing style augmentation methods—typically based on nonlinear transformations—fail to accurately capture the target domain distribution. Moreover, most DG approaches neglect the issue of negative knowledge transfer, leading to degraded performance in the source domain. To address these challenges, we propose a structure-constrained, diffusion-based style divergence augmentation that operates in the frequency domain using a continuous style combination mechanism. This generates diverse samples with broad domain coverage, improving representation robustness. Furthermore, inspired by biological genetics, we introduce DNA-Net, a genetic-inspired dual-chain collaborative learning framework. By jointly optimizing two related tasks—source domain image reconstruction and generalized segmentation—DNA-Net explicitly suppresses negative transfer while enhancing cross-domain segmentation performance. Extensive experiments on two public medical image benchmarks demonstrate that our approach surpasses state-of-the-art DG methods, achieving superior performance on both source and target domains. Our code is available at `https://anonymous.4open.science/r/DNA-Net-SESD-5891/`.

## 1 Introduction

Domain Generalization (DG) in medical image segmentation is a significant challenge. It requires models trained on data from the source domain (e.g., a specific hospital or imaging device) to effectively segment images across diverse distributions, including those from various hospitals, devices, or patient populations Li et al. (2020; 2021b;c). This generalization capability is essential for real-world applications, as medical imaging data often display high variability and complexity. Inter-source differences can lead to overfitting on training data, degrading model performance on previously unseen domains.

To address this issue, most existing methods focus on data augmentation and model training strategies Su et al. (2023); Li et al. (2024); Cheng et al. (2023); Robey et al. (2021); Liu et al. (2021); Robey et al. (2021). While these approaches have achieved notable success in domain generalization, two key challenges remain: (1) Current data augmentation techniques often struggle to sufficiently enhance the diversity and information richness of augmented samples, limiting their ability to comprehensively cover potential target domain distributions. (2) Many domain adaptation strategies aim to minimize distributional discrepancies between the source and target domains; however, this can lead to negative transfer, where knowledge beneficial for the target domain is transferred at the expense of performance on the source domain (We term inter-domain negative transfer). These critical issues motivate us to explore methods for expanding target domain coverage in medical image segmentation while avoiding inter-domain negative transfer, thereby preserving performance in the source domain.

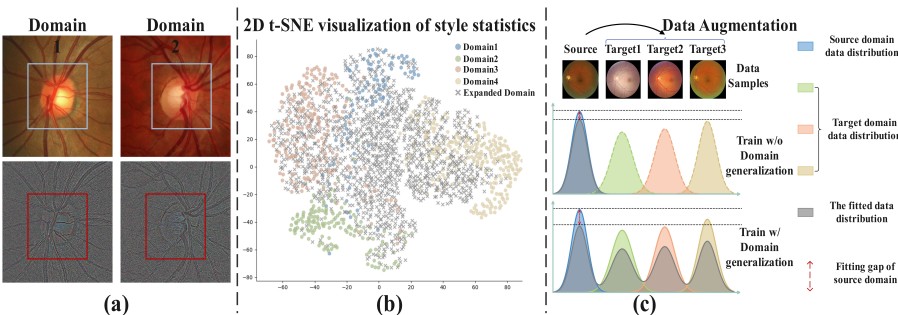

Figure 1: Conceptual overview of our motivation. (a) Original and phase spectrum images of fundus images acquired from different devices. (b) Coverage of samples from unseen domains after applying traditional data augmentation and diffusion generation methods in the source domain. (c) Performance of the model on the source domain without domain generalization compared to domain generalization with data augmentation.

Recently, diffusion models have attracted significant attention due to their strong modality coverage and high-quality sample generation, with increasing applications in medical imaging Carlini et al. (2023); Cao et al. (2024); Croitoru et al. (2023). Compared to traditional augmentation techniques, diffusion-based generation methods offer distinct advantages: Diffusion generation methods excel at preserving critical details during image enhancement, effectively avoiding information loss that often results from color transformations. Additionally, these methods are capable of generating entirely new samples and are better equipped to handle the complex distributions of medical image data, leading to more precise enhancement outcomes.

Therefore, we performed a statistical analysis of feature distributions across data generated by various augmentation methods, revealing the following insights: (1) Cross-domain data exhibit anatomical consistency, with differences primarily in contrast, texture, and brightness (i.e., style inconsistencies). For instance, as shown in Figure 1(a), fundus images from different devices share similar anatomical structures, with cross-domain variation largely reflected in contrast and brightness. (2) Combining traditional augmentation with diffusion generation techniques better covers cross-domain data distributions, thus enhancing domain adaptation. In Figure 1(b), diffusion generation atop traditional augmentation yields the expanded domain with richer style information. (3) Data augmentation improves model generalization in domain generalization tasks, allowing better adaptation to cross-domain feature distributions. However, it may also induce inter-domain negative transfer, degrading performance on the source domain. Figure 1(c) illustrates that while a model trained on source-only data captures source domain features effectively, data augmentation introduces a more complex distribution that can impair source domain performance.

Our approach has clinical significance. High performance in source domain preserves model effectiveness in known environments and boosts credibility in real-world apps. A model trained on one hospital's data can be applied to others Ma et al. (2024a); Li et al. (2021a); Wang et al. (2023). Declining source domain performance risks missed diagnoses and delays in treatment for original hospital patients. Safeguarding it ensures diagnostic accuracy for original patient population and enhances clinical safety when deployed across hospitals.

Based on the aforementioned motivations, we hypothesize that in medical scenarios, there is anatomical structure consistency and style distribution disparity between the source and target domains. Building on this, we introduce a structure-guided style augmentation module. Specifically, we apply traditional style augmentation techniques Zhou et al. (2022c) to source domain data to capture broader domain style information, thereby creating an extended domain. We then perform a Fourier transform to extract frequency-domain signals from the expanded domain images. The amplitude and phase spectra in the frequency domain are used to capture the style distribution and anatomical structure, respectively. The amplitude spectrum (representing style) undergoes further diffusion generation, which is then recombined with the phase spectrum to produce samples that cover a

broader feature distribution. Our frequency-domain diffusion approach mitigates the high computational cost and memory demands associated with performing direct diffusion generation on medical images. Building on this, inspired by the complex mechanisms of biological genetics, we propose a collaborative learning network modeled after the structure of DNA-like dual-chains (DNA-Net). This network facilitates collaborative learning through two related tasks: source domain image restoration and generalization segmentation, addressing the issue of inter-domain negative transfer during generalization. Additionally, drawing from natural selection principles Zheng et al. (2020), we design a novel loss function for DNA-Net, employing a strong-weak natural selection strategy to guide the network's learning bias. This strategy directs task selection and inter-layer knowledge transfer within the network, enhancing both the network's stability and the accuracy of generalized segmentation. The main contributions of this paper are summarized as follows:

- We apply amplitude spectrum diffusion to data augmentation for medical image domain generalization. We address the issues of poor feature coverage across different domains in traditional data augmentation methods and the high resource demands associated with directly performing diffusion generation on medical images.

- We introduce a new perspective for building domain generalization models, inspired by the complex mechanisms of biological genetics. From this novel perspective, we propose a DNA-like dual-chain structure collaborative learning network and a novel learning bias-guided loss function, effectively addressing the previously unresolved issue of inter-domain knowledge negative transfer in domain generalization research.

- Experimental results demonstrate that our method outperforms existing SOTA approaches on both single-source and multi-source domain generalization datasets. Additionally, it achieves superior segmentation performance on source domain data compared to current SOTA methods.

## 2 Related Work

### 2.1 Medical Image Diffusion Generation

Diffusion generation in medical imaging simulates the inherent distribution characteristics of medical image data to generate diverse and realistic samples, thereby enhancing the model's ability to generalize across different imaging conditions and pathological features. Recent studies Xing et al. (2024) have proposed a cross-conditioned diffusion model for medical image-to-image translation, where the source modality is used as input, and the target modality is synthesized under the guidance of the target distribution. Zhan et al. Zhan et al. (2024) introduced adaptive cross-guided parameter adjustments within a multi-stream diffusion framework and proposed a medical multimodal generation approach. Liu et al. Liu et al. (2023) alternated between data fidelity and sampling updates in a transformation sine graph-based diffusion model to synthesize high-quality, structurally consistent image data. Graikos et al. Graikos et al. (2024) trained a diffusion model based on self-supervised learning embeddings for generating high-quality large images. A key challenge with these methods is the need to preserve critical details, such as anatomical boundaries and pathological features, in the original medical images when generating new samples. Computational limitations can slow sampling speeds, potentially introducing blurring or noise, which can lead to severe distortions in the generated images. In contrast, we propose a novel approach that focuses on amplitude spectrum diffusion, generating only structurally simple style information while preserving anatomical details (phase spectrum). This method avoids the high computational costs and reliability issues associated with direct diffusion generation on medical images.

### 2.2 Domain Generalization

The goal of domain generalization is to enhance the model's ability to generalize to unseen domains, addressing the challenges posed by varying data distributions Zhou et al. (2022a); Wang et al. (2021); Zhao et al. (2019); Ma et al. (2024b); Chen et al. (2023). This presents a common yet challenging task. In recent years, DG has garnered significant attention from researchers. Choi et al. Choi et al. (2023) proposed a progressive random convolution method that creates more effective virtual domains by gradually increasing style diversity. Pei et al. Pei et al. (2024) adopted a multi-level adversarial learning scheme to adapt to different levels of features between each source domain and

the target domain, aiming to enhance segmentation performance. Zhou et al. Zhou et al. (2022c) introduced a dual normalization model that uses augmented source-similar and source-dissimilar images to address generalization tasks in cross-modal settings. While these methods have achieved promising results in domain generalization research, they overlook the issue of negative transfer of domain knowledge when mapping source domain data to target domains. Our work specifically addresses this unresolved problem of negative knowledge transfer in domain generalization. We focus on ensuring that the model not only generalizes well to unseen domains but also maintains high performance on source domain data, thus preventing performance degradation.

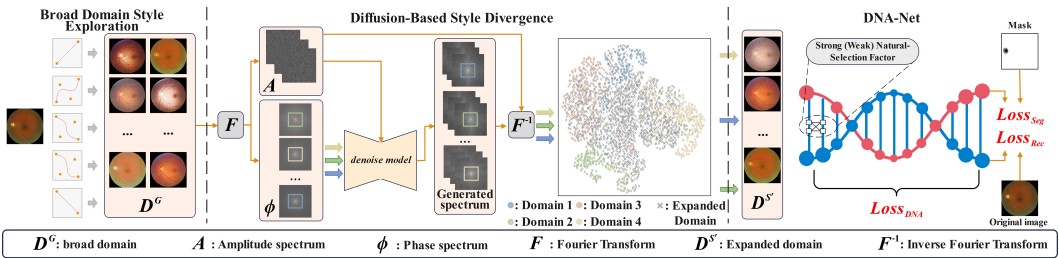

Figure 2: The overall workflow of our method. First, we employ a broad domain style exploration module to obtain a broad domain $D^G$. Next, we expand the data samples using our proposed diffusion-based style divergence method. Finally, the generated data is fed into our proposed DNA-Net for training.

## 3 METHODOLOGY

### 3.1 DEFINITION AND OVERVIEW

In a domain generalization task, there is source domain $D^S$ and unknown target domain $D^T$. The source domain $D^S = \left\{ x_i^S, y_i^S \right\}_{i=1}^{N^S}$, where $S$ represents the source domain name, $D_i^S$ refers to the $i$-th sample in the source domain, and $y_i^S$ represents the corresponding segmentation mask for that sample. $N^S$ is the total number of samples. Our goal is to train a segmentation model $S_{\text{seg}}$ using only samples from $D^S$ and ensure its ability to generalize to unseen target domains. As shown in Figure 2, we first perform broad domain style exploration on $D^S$ using a style divergence module based on structural constraints, obtaining a broad domain $D^G$. Then, we perform further divergence on $D^G$ to obtain an expanded domain $D^{S'} = \left\{ x_i^{S'}, y_i^{S'} \right\}_{i=1}^{N^{S'}}$. Due to the structural similarity between the source and target domains, the style enhancement guided by structure allows $D^{S'}$ to incorporate both source and target domain feature distributions. Based on this, we introduce a collaborative learning network (DNA-Net). One chain trains the segmentation model $S_{\text{seg}} = x^{S'} \to y^{S'}$, while the other chain trains the reconstruction model $S_{\text{rec}} = x^{S'} \to x^S$. The $S_{rec}$ learns the mapping from $D^S$ to $D^{S'}$, and through the collaborative network, feature-level knowledge is transferred to $S_{\text{seg}}$, allowing it to learn feature-level mappings from $D^S$ to $D^{S'}$. This ensures the model's adaptability to both the source domain and unseen target domains.

### 3.2 STRUCTURE-GUIDED STYLE AUGMENTATION

#### 3.2.1 BROAD DOMAIN STYLE EXPLORATION

For generalizable image segmentation tasks, style differences across domains primarily manifest as variations in global or localized brightness, contrast, and subtle color shifts Bi et al. (2024); Kang et al. (2023); Spanos et al. (2024). Based on this, we designed a broad-domain style exploration module that employs pixel-level nonlinear transformations to explore styles beyond the $D^S$, resulting in $D^G$. Specifically, we selectively extract style features based on image characteristics in different generalization tasks and apply pixel-level mappings. Using a monotonic nonlinear transformation

function, we map the pixel-level style features of the original image to new values, generating images with diverse styles and enabling exploration from $D^S$ to $\{D_1^T, ..., D_n^T\}$. To achieve this, Following Zhou et al. (2022c), we utilize Bézier curves for pixel-level nonlinear transformations.

$$B(t) = \sum_{i=0}^{n} \binom{n}{i} t^i (1-t)^{n-i} P_i, \cdots 0 \le t \le 1 \tag{1}$$

where, $B(t)$ represents the output of the Bézier curve equation at a given parameter $t$, indicating the coordinates of the point on the curve corresponding to $t$. The parameter $n$ denotes the degree of the Bézier curve, and $p_i$ represents the $i$-th control point, which determines the curve's shape. The binomial coefficient $\binom{n}{i}$ is the weight factor for the $i$-th control point in the Bézier equation.

### 3.2.2 DIFFUSION-BASED STYLE DIVERGENCE

To further enhance sample diversity and improve the coverage of generated samples in the target domain, we propose a diffusion-based style divergence method. This method utilizes the fixed image contours and structural information in the phase spectrum Xu et al. (2021), while leveraging the diffusion model's powerful capability in learning data distributions and generating samples. The resulting style-diverse samples preserve anatomical structure consistency, allowing the style of $D^G$ to fully diverge and yielding $D^{S'}$.

**Forward process:** First, we extract the magnitude spectrum $A(u,v)$ and phase spectrum $\Phi(u,v)$ of the target image using a Fourier transform Xu et al. (2021), where $(u,v)$ represents the frequency coordinates. Next, we employ the phase spectrum as a constraint to guide the diffusion model in aligning the feature distribution of the magnitude spectrum. This separation of the two components alleviates the computational load on the model while maintaining structural consistency in the generated images. Specifically, during the forward noise addition process, we degrade the magnitude spectrum by introducing Gaussian noise, as described by the following equation:

$$q(x_t \mid x_0) = N\left(x_t; \sqrt{\bar{\alpha}_t} \cdot x_0, (1 - \bar{\alpha}_t) \cdot I\right) \tag{2}$$

where, $x_0$ represents the original data, and $x_t$ is the noisy data at time step $t$. The cumulative attenuation coefficient from step 1 to step $t$ is denoted as $\bar{\alpha}_t = \prod_{s=1}^{t} \alpha_s$, where $\alpha_t = 1 - \beta_t$ and $\beta_t$ represents the noise intensity at each time step. $N(\cdot; u, \sigma^2)$ denotes a normal distribution with mean $u$ and variance $\sigma^2$.

**Reverse process:** The goal of the reverse denoising process is to recover the original data $x_0$ from the noisy data $x_t$, using the phase spectrum information $\Phi(u,v)$ as a constraint. The specific formula is as follows:

$$p_\theta(x_{t-1} \mid x_t, \Phi) = N(x_{t-1}; u_\theta(x_t, t, \Phi), \Sigma_\theta(x_t, t)) \tag{3}$$

where, $\Sigma_\theta(x_t, t)$ is the variance term, representing the uncertainty in generating the sample $x_{t-1}$. The parameter $\theta$ is optimized through training. $u_\theta(x_t, t, \Phi)$ is the mean prediction function dependent on the conditional information $\Phi$, representing the prediction of $x_{t-1}$ from $x_t$ at time step $t$. The specific formula is as follows:

$$u_\theta(x_t, t, \Phi) = \frac{1}{\sqrt{\alpha_t}}\left(x_t - \frac{\beta_t}{\sqrt{1 - \bar{\alpha}_t}} \epsilon_\theta(x_t, t, \Phi)\right) \tag{4}$$

where, $\epsilon_\theta(x_t, t, \Phi)$ represents the noise component predicted by the model. The model fits the noise $\epsilon$ added during the forward process based on $x_t$, the time step $t$, and the condition $\Phi$.

**Sample process:** The integrity of anatomical structures is crucial in medical imaging, as any deviation or missing structure can significantly impact the reliability of the image Li et al. (2023b); Hu et al. (2022a). We analyze the differences in the frequency domain of images across common medical imaging generalization tasks and find that these differences primarily manifest in the magnitude spectrum. Meanwhile, previous studies have shown that the phase spectrum retains contour and structural information Xu et al. (2021); Liu et al. (2024). Based on this, we propose a continuous combination mechanism in the diffusion frequency space to generate diverse and reliable samples. Specifically, we first extract the phase and magnitude spectra of the original image using Fourier transform. Then, we sample initial noise $x_t'$ from a standard normal distribution. Next, the phase spectrum is fed as a condition into the noise prediction model to constrain the solution space, and we use Denoising Diffusion Implicit Model for fast sampling. Finally, the generated magnitude spectrum $A'(u,v)$ is combined with the original phase spectrum $\Phi(u,v)$, and an inverse Fourier transform is applied to complete the divergence of the $D^G$ image.

## 3.3 DNA-NET

The key to transferring knowledge from the source domain to an unknown domain lies in establishing a mapping relationship between different visual domains, enabling effective knowledge transfer from the source to the unknown domain Li et al. (2023b). The features of a visual domain are typically represented by the image style, which is primarily captured in the statistical features of shallow CNN layers Huang & Belongie (2017). Therefore, we propose that, during the reconstruction of the model $S_{rec}$ to restore the extended domain $D^{S'}$ to the source domain $D^S$, the intermediate layers contain rich style mapping knowledge. This knowledge can assist the segmentation model $S_{seg}$ in understanding the relationship between the source and extended domains, thus mitigating negative transfer from the source domain and improving the model's ability to adapt to the mappable unknown domain.

Based on this, we propose a collaborative learning network with a DNA-like dual-chain structure (DNA-Net), as shown in Figure 3. The network captures mapping information indirectly through collaborative learning, providing implicit representations of the source and target domain distributions Yuan et al. (2022); He et al. (2021). Inspired by biological genetics and natural selection theories, we design a feature-learning bias-guided strategy based on strong-weak natural selection to adjust the learning loss of DNA-Net. Through local and global bias-guided adjustments, we aim to enhance the model's robustness and stability (See the supplementary material for detailed examples).

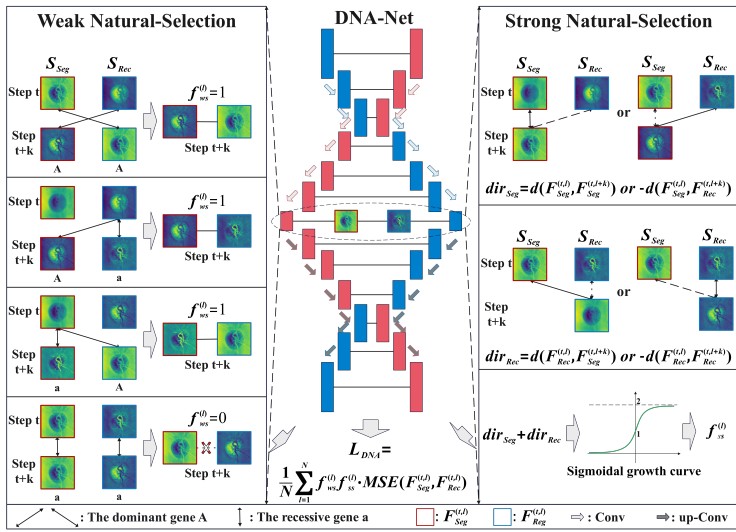

Figure 3: The overall architecture of our proposed DNA-Net, illustrating the strong and weak natural selection strategies within the network (for visual clarity, skip connections within each U-Net chain are omitted).

### 3.3.1 WEAK NATURAL-SELECTION STRATEGY

In nature, individuals of the same species exhibit trait diversity (e.g., eye color, single or double eyelids), which is primarily driven by genetic factors rather than natural selection. These trait differences arise mainly due to genetic variations, a phenomenon known as weak natural selection Zheng et al. (2020).

Inspired by this concept, we introduced the notion of "genes" to enhance the diversity and selectivity of learning in the intermediate layers of DNA-Net, proposing a weak-selection-based local bias strategy. Specifically, we computed the similarity of intermediate layer $l$ features from both the segmentation and reconstruction models at different training stages. By comparing the Euclidean distance between layer feature similarities, we derived the network's "genes." (Where, in training steps $t$ and $t + k$, if the feature map similarity between different tasks is minimal, we define it as the dominant gene $A$; if the feature map similarity within the same task is minimal, we define

it as the recessive gene $a$.) We then assigned different weak natural selection factors $f_{ws}^{(l)}$ to the "traits" exhibited by various gene combinations, thereby controlling the inter-layer learning bias of the model (As shown in Appendix Algorithm 1).

It is important to note that, to prevent the model from performing prolonged independent learning under recessive traits (where traits remain recessive for an extended period Proudfoot et al. (1982)), we designed a gene mutation-based coercive collaboration mechanism. This mechanism introduces a minimal probability $\delta$ of gene reversal, thereby encouraging collaborative learning among the model layers.

### 3.3.2 STRONG NATURAL-SELECTION STRATEGY

In nature, species in different environments often exhibit significant differences, primarily influenced by their surroundings. Only populations that successfully adapt to the environment can survive, a process known as strong natural selection Zheng et al. (2020). Strong natural selection enhances the robustness and stability of populations, playing a crucial role in the successful evolution of a species.

Similarly, to enhance the stability of the collaborative learning network and improve both the model's performance and convergence speed, we propose a global bias strategy based on strong natural selection. Given that the ultimate goal is the segmentation task, we treat the "environment" as being favorable for segmentation, thereby biasing the model towards this task. During the collaborative training process, we apply a global task bias using the strong natural selection factor $f_{ss}^{(l)}$, encouraging the model to learn more effectively from the segmentation model (As shown in Appendix Algorithm 2).

The above method allows DNA-Net to focus on efficiently completing the segmentation task.

### 3.4 LOSS FUNCTION

To enhance the collaborative learning capability of DNA-Net and improve the stability of the dual-chain model, we propose a novel loss function $L_{DNA}$. This loss function utilizes mean squared error to measure the difference between intermediate layer feature maps, adjusting it with the weak natural selection factor $f_{ws}^{(l)}$ and the strong natural selection factor $f_{ss}^{(l)}$. The formula is as follows:

$$L_{DNA} = \frac{1}{N} \sum_{l=1}^{N} f_{ws}^{(l)} \cdot f_{ss}^{(l)} \cdot \frac{1}{M} \sum_{i=1}^{M} \left( F_{seg,i}^{(t,l)} - F_{rec,i}^{(t,l)} \right)^2 \tag{5}$$

where, $N$ is the number of intermediate layers, $M$ is the number of pixels in each feature map, $f_{ws}^{(l)}$ and $f_{ss}^{(l)}$ are the weight adjustment factors for the $l$-th layer, and $F_{seg,i}^{(t,l)}$ and $F_{rec,i}^{(t,l)}$ represent the $i$-th feature value at the $l$-th layer of the segmentation model and reconstruction model, respectively, at the $t$-th stage.

For the total loss of the segmentation model, we combine the cross-entropy loss, Dice loss, and $L_{DNA}$ to enhance gradient stability and the model's pixel-level segmentation capability. The formula is as follows:

$$L_{seg} = \alpha \cdot L_{CrossEntropy} + \beta \cdot L_{Dice} + \gamma \cdot L_{DNA} \tag{6}$$

where, $\alpha$, $\beta$, and $\gamma$ represent the mixing weights for the three losses. The total loss of the reconstruction model is similar, combining the structural similarity loss $L_{SSIM}$ and $L_{DNA}$, and is given by the following formula:

$$L_{rec} = \alpha' \cdot L_{SSIM} + \beta' \cdot L_{DNA} \tag{7}$$

## 4 EXPERIMENTS

### 4.1 EXPERIMENTAL SETTING

**Datasets and Preprocessing.** We conducted experiments on two different medical image domain generalization datasets, covering both single-source and multi-source domain generalization, to

demonstrate the effectiveness and generalizability of our method: the BraTS dataset Menze et al. (2014) and the fundus image dataset Orlando et al. (2020). The BraTS dataset consists of 210 high-grade glioma cases and 75 low-grade glioma cases, with each case comprising four MRI modalities: T2, Flair, T1, and T1CE. The fundus image dataset includes data from four domains collected using different scanners at various institutions, with the primary task being the segmentation of the optic cup and optic disc in fundus images.

**Implementation Details.** All experiments were conducted in a Python 3.9 and PyTorch 2.4 environment on an Ubuntu 22.04 system, utilizing 5 NVIDIA RTX 4090 GPUs. For data preprocessing, we sliced the 3D images of the BraTS dataset along the scanning direction, resized them to 240×240, normalized the values to the range [-1,1], and split them into training, validation, and test sets with a 7:1:2 ratio. For the fundus image dataset, we scaled the images from 800×800 to 384×384 and used the provided train-test splits. For model training, we set the number of epochs to 100 and the batch size to 8, using the Adam optimizer with an initial learning rate of 0.0001. Additionally, our diffusion model was configured with 1000 denoising steps, an initial noise variance of 0.0001, a maximum variance of 0.02, and linear noise scheduling. Both DNA strands in the DNA-Net model use U-Net as the backbone network, with $L_{DNA}$ decayed every 200 steps by a factor of 0.5, and the gene mutation probability $\delta$ set to 0.05. The loss function parameters $\alpha, \beta, \gamma, \alpha', \beta'$ are set to 0.25, 0.25, 0.5, 0.5, and 0.5, respectively. We use the Dice similarity coefficient, Average Surface Distance (ASD) and Hausdorff Distance (HD) to evaluate segmentation performance.

Table 1: The comparison results of our method with other related works on the BraTS dataset (including using T2 as the source domain and T1CE as the source domain) are presented below.

| Method | Source Domain: T2 | | | | | | | | Source Domain: T1CE | | | | | | | |
|---|---|---|---|---|---|---|---|---|---|---|---|---|---|---|---|---|
| | Dice↑ | | | | HD↓ | | | | Dice↑ | | | | HD↓ | | | |
| | Flair | T1 | T1CE | Average | Flair | T1 | T1CE | Average | Flair | T1 | T2 | Average | Flair | T1 | T2 | Average |
| No Generalization | 73.37 | 5.02 | 10.69 | 29.69 | 11.96 | 38.65 | 37.44 | 29.35 | 46.64 | 60.26 | 15.67 | 40.85 | 21.56 | 18.03 | 47.76 | 29.12 |
| Fed-DG Liu et al. (2021) | 75.77 | 5.82 | 9.51 | 30.37 | 14.45 | 54.03 | 51.06 | 39.85 | 33.03 | 58.30 | 4.09 | 31.72 | 32.07 | 22.35 | 56.08 | 36.83 |
| MixStyle Zhou et al. (2021) | 77.03 | 45.68 | 40.23 | 54.31 | 12.97 | 23.10 | 24.36 | 20.14 | 37.55 | 63.12 | 68.31 | 56.32 | 28.75 | 18.74 | 14.91 | 20.80 |
| CSDG Ouyang et al. (2022) | 61.37 | 47.53 | 43.84 | 50.91 | 16.74 | 22.77 | 21.58 | 20.36 | 42.11 | 62.77 | 65.79 | 56.89 | 22.15 | 19.75 | 16.73 | 19.54 |
| SADN Zhou et al. (2022c) | 75.87 | 49.36 | 38.09 | 54.44 | 13.44 | 20.15 | 23.56 | 19.05 | 47.31 | 63.64 | 63.00 | 57.98 | 21.03 | 18.06 | 17.56 | 18.88 |
| EGSDG Jiang & Gu (2024) | 76.16 | 62.43 | 55.87 | 64.82 | 13.43 | 18.84 | 17.79 | 16.69 | 68.49 | 71.22 | **73.06** | 70.92 | 17.81 | 12.93 | **14.74** | 15.16 |
| **DNA-Net** | **81.21** | **66.54** | **66.73** | **71.49** | **8.60** | **13.03** | **13.93** | **11.85** | **72.95** | **72.02** | 68.47 | **71.15** | **10.34** | **10.87** | 14.81 | **12.04** |

Table 2: Comparison of federated domain generalization results on Optic Disc/Cup segmentation from fundus images (We follow the practice in domain generalization literature to adopt the leave-one-domain-out strategy).

| Method | Target Domain: Domain1 | | | | Target Domain: Domain2 | | | | Target Domain: Domain3 | | | | Target Domain: Domain4 | | | |
|---|---|---|---|---|---|---|---|---|---|---|---|---|---|---|---|---|
| | Disc | | Cup | | Disc | | Cup | | Disc | | Cup | | Disc | | Cup | |
| | Dice | ASD | Dice | ASD | Dice | ASD | Dice | ASD | Dice | ASD | Dice | ASD | Dice | ASD | Dice | ASD |
| No Generalization | 82.80 | 19.03 | 64.55 | 26.20 | 85.99 | 19.98 | 76.03 | 20.31 | 88.78 | 16.63 | 84.14 | 10.35 | 85.43 | 9.93 | 68.85 | 25.11 |
| Fed-DG Liu et al. (2021) | 95.47 | 7.81 | 81.66 | 18.79 | 86.34 | 19.57 | 76.31 | 19.98 | 93.36 | 9.12 | 85.23 | 10.86 | 94.68 | 6.02 | 85.27 | 8.94 |
| DoFE Wang et al. (2020) | 96.04 | 7.05 | 81.95 | 18.59 | 89.20 | 15.75 | 78.31 | 16.40 | 93.23 | 9.76 | 85.51 | 10.06 | 94.28 | 6.99 | 86.61 | 8.28 |
| RAM-DSIR Zhou et al. (2022b) | 95.75 | 7.12 | 85.48 | 16.05 | 89.43 | 13.86 | 78.82 | 14.01 | 94.67 | 7.11 | 87.44 | 9.02 | 94.10 | 7.06 | 85.84 | 8.29 |
| DCAC Hu et al. (2022b) | 96.52 | 6.35 | 81.43 | 19.20 | 87.85 | 18.28 | 77.72 | 17.15 | 94.28 | 8.11 | 86.80 | 9.14 | **95.40** | 5.20 | 87.68 | 7.12 |
| DFQ Bi et al. (2024) | 96.50 | **6.01** | **87.30** | 15.72 | 92.52 | 12.09 | 81.92 | 13.05 | 95.04 | 7.05 | 88.95 | 7.70 | 94.85 | 5.84 | 87.47 | 6.55 |
| DNA-Net | **96.53** | 6.15 | 86.58 | **15.35** | **92.71** | **11.87** | **83.79** | **11.76** | **95.11** | **6.98** | **90.02** | **7.38** | 95.28 | **5.17** | **88.53** | **6.38** |

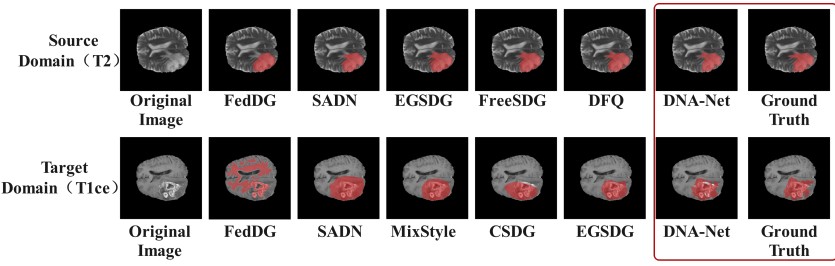

Figure 4: Visual comparison of model prediction results on T2 source domain and T1ce target domain.

## 4.2 COMPARISON WITH STATE-OF-THE-ART METHODS

To evaluate the performance of our method more comprehensively, we present the results in Table 1, which shows the performance on the BraTS dataset, assessing the single-source domain generaliza-

tion capability of our approach. In Table 2, we report the results on the Fundus dataset, evaluating its performance on the multi-source domain generalization task. We compare our proposed method with SOTA methods to highlight its effectiveness. The results demonstrate that our method achieves excellent generalization performance in both single-source and multi-source domain generalization tasks.

Additionally, to further validate the protection ability of DNA-Net on source domain performance, we tested the generalization results of the comparison methods on source domain data. The experimental results are shown in Table 3. We also visualized and compared the prediction results of the model in the source domain and the target domain. (The results of the Fundus dataset are shown in Table 6 of the appendix)

Table 3: The source domain performance of the model on the BraTS dataset.(Baseline means training and testing only in the source domain)

| Method | Source Domain: T2 | | Source Domain: T1CE | |
|---|---|---|---|---|
| | Dice | HD | Dice | HD |
| Baseline (No Generalization) | 84.52 | 4.47 | 78.71 | 6.63 |
| Fed-DG Liu et al. (2021) | 68.68 | 19.21 | 59.32 | 23.52 |
| FreeSDG Li et al. (2023a) | 77.63 | 11.84 | 72.11 | 12.01 |
| SADN Zhou et al. (2022c) | 73.71 | 18.45 | 55.15 | 27.63 |
| EGSDG Jiang & Gu (2024) | 75.93 | 13.61 | 63.67 | 18.37 |
| DFQ Bi et al. (2024) | 78.11 | 12.13 | 70.41 | 13.86 |
| **DNA-Net** | **84.89** | **4.58** | **78.21** | **7.21** |

The results indicate that, compared to the baseline trained exclusively on source domain data, existing domain generalization methods generally exhibit decreased performance on source domain data. In contrast, our proposed DNA-Net effectively mitigates the issue of negative knowledge transfer between domains. This improvement is attributed to DNA-Net's collaborative learning and bias-guided strategy, which jointly optimize source reconstruction and segmentation to preserve knowledge and reduce negative transfer.

## 4.3 ABLATION STUDY

We conducted an ablation study to evaluate the key components of our method, including Broad Domain Style Exploration (SE), Diffusion-Based Style Divergence (SD), Weak Natural Selection Strategy (WS), and Strong Natural Selection Strategy (SS). The full method achieved the best performance, demonstrating the indispensability of each module (As shown in table 4).

Table 4: The source domain performance of the model on the BraTS dataset.

| Method | SE | SD | WS | SS | T2(Source) | | T2-other(Avg) | | T1CE(Source) | | T1CE-other(Avg) | |
|---|---|---|---|---|---|---|---|---|---|---|---|---|
| | | | | | Dice↑ | HD↓ | Dice↑ | HD↓ | Dice↑ | HD↓ | Dice↑ | HD↓ |
| VA1 | - | - | - | - | 84.52 | 4.47 | 29.69 | 29.35 | 78.71 | 6.63 | 40.85 | 29.12 |
| VA2 | ✓ | - | - | - | 80.56 | 7.39 | 51.82 | 19.76 | 74.98 | 9.69 | 44.30 | 29.78 |
| VA3 | - | ✓ | - | - | 83.16 | 5.39 | 33.35 | 29.88 | 76.42 | 7.28 | 44.13 | 20.40 |
| VA4 | ✓ | ✓ | - | - | 79.33 | 6.51 | 67.59 | 12.04 | 72.83 | 9.20 | 61.71 | 15.11 |
| VA5 | ✓ | ✓ | ✓ | - | 84.40 | 5.38 | 65.99 | 15.54 | 78.19 | 7.07 | 69.39 | 12.53 |
| VA6 | ✓ | ✓ | - | ✓ | 81.37 | 7.07 | 58.97 | 19.57 | 77.47 | 8.06 | 59.48 | 16.99 |
| **Ours** | ✓ | ✓ | ✓ | ✓ | **84.89** | 4.58 | **71.49** | **11.85** | 78.21 | 7.21 | **71.15** | **12.04** |

When compared to the model trained solely on source domain data (VA1), the performance on the target domain improved with the application of the structure-guided style augmentation methods (VA2, VA3, VA4), although the corresponding performance on the source domain declined. This result further supports the rationale behind our approach. Notably, when only SD is applied to the source domain data, the performance decline on the source domain is minimal, while the improvement on the target domain remains limited. This behavior highlights that, in the absence of style exploration, the model can only diffuse data from the source domain distribution, without generating samples from other domains, underscoring the necessity of SE. Furthermore, as demonstrated in VA5 and VA6, the feature-learning bias-guided strategy in DNA-Net plays a crucial role in preserving source domain performance. VA5 provides better source domain protection than VA6, and models incorporating both WS and SS achieve the best source domain protection. From a biological genetics perspective, weak natural selection tends to slow evolutionary progress, preserving stability within the original ecosystem (i.e., the source domain). However, an ecosystem incorporating both selection strategies exhibits greater overall stability.

## 5 CONCLUSION

We propose a novel data augmentation approach based on amplitude spectrum diffusion. Drawing inspiration from the complex mechanisms of biological genetics, we introduce a DNA-like dual-chain collaborative learning network, along with an innovative bias-guided loss function, to address the unresolved problem of negative inter-domain knowledge transfer in domain generalization studies. Experimental results show that the proposed method performs well on both single-source and multi-source domain generalization datasets, and achieves better segmentation performance on source domain data compared to current SOTA methods.

## 6 ETHICS STATEMENT

This study uses only publicly available, de-identified medical imaging datasets released under ethical approval by their original providers. No personally identifiable information or private patient data was collected, processed, or disclosed by the authors. All experiments comply with relevant privacy, security, and legal requirements, and follow the principles outlined in the ICLR Code of Ethics. The proposed DNA-Net framework is intended solely for research and educational purposes, with the aim of improving domain generalization in medical image segmentation without introducing harmful bias or discriminatory outcomes.

## 7 REPRODUCIBILITY STATEMENT

To ensure the reproducibility of our results, we provide an anonymous, complete implementation of DNA-Net, including all training scripts, model configurations, and pre-processing steps, at `https://anonymous.4open.science/r/DNA-Net-SESD-5891/`.

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

## A APPENDIX

## B THE USE OF LARGE LANGUAGE MODELS (LLMS)

Large Language Models (LLMs) were used in this work solely as a general-purpose assist tool for language refinement, grammar correction, and improving the clarity of the manuscript. LLMs were not involved in research ideation, dataset design, experimental implementation, or result generation. All technical content, scientific claims, and conclusions are entirely the authors' original work. The authors take full responsibility for the accuracy and integrity of the final manuscript.

## C BEZIER VISUALIZATION RESULTS

We visualized the results of the wide-area style exploration module. The core of broad domain style exploration is the Bessel transform. Figure 5 shows the original fundus and brain images along with their corresponding Bezier transformed images. From these images, we observe that the generated images encompass a broader range of styles.

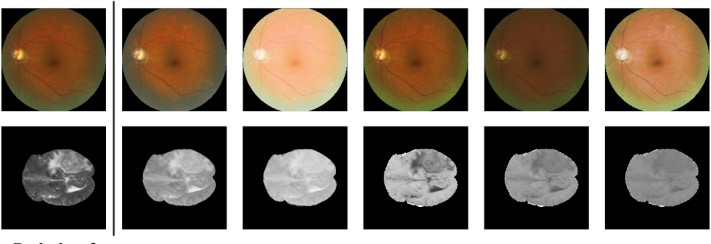

**Original image**   **The image after applying the Bézier transform.**

Figure 5: Original fundus and brain images and their corresponding Bezier transformed images.

## D DIFFUSION GENERATION RESULTS DISPLAY

### D.1 AMPLITUDE SPECTRUM RESULTS OF DIFFUSION GENERATION

We extracted the corresponding phase and magnitude spectrum information from the BraTS and Fundus datasets, respectively. Then, we used our proposed structure-guided style diffusion method to generate new magnitude spectrum images. The generated results are shown in Figure 6.

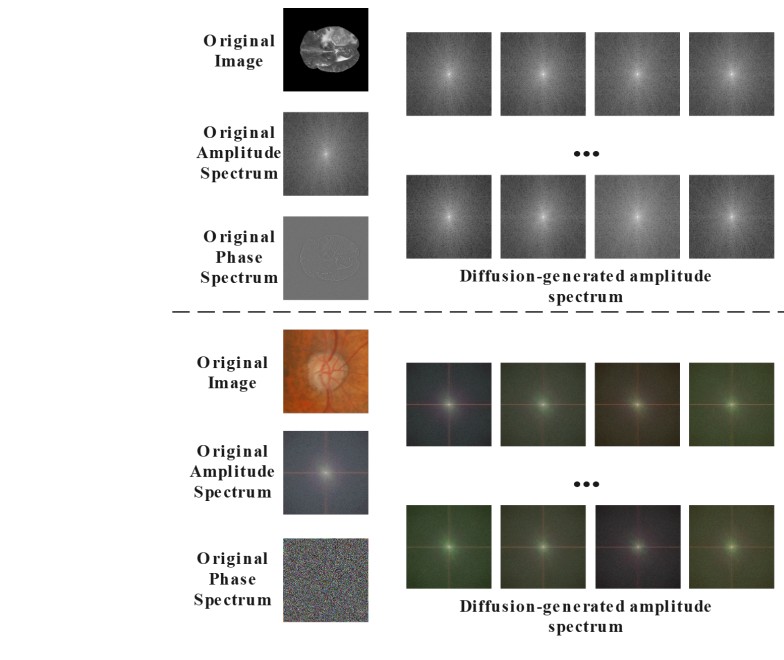

Figure 6: Generate new amplitude spectrum images using our proposed structure-guided diffusion method.

## D.2 THE FINAL GENERATED MEDICAL IMAGING RESULTS

We combine the generated amplitude spectrum information with the original phase spectrum of the image, and then perform an inverse Fourier transform to obtain a sample with new style information, as shown in Figure 7.

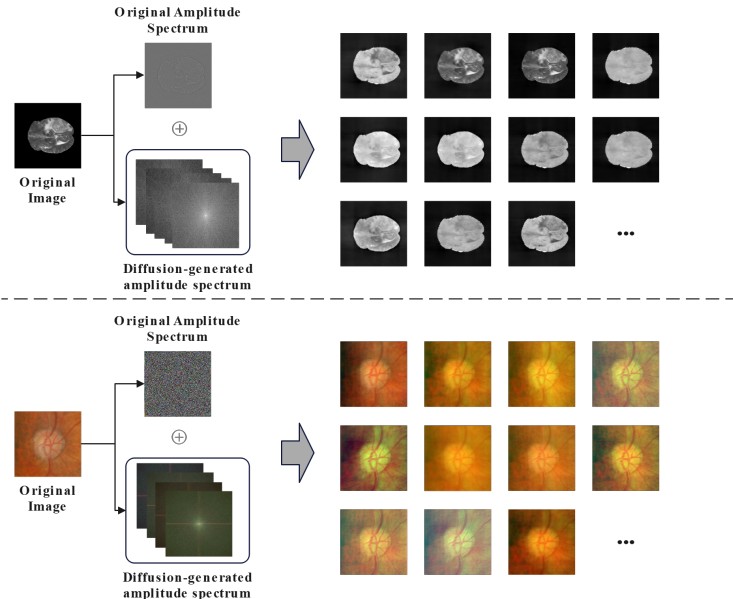

Figure 7: New samples obtained by combining the original phase spectrum with the generated amplitude spectrum.

It is worth noting that, compared to directly applying diffusion on the original images, our method has two key advantages:

- We use the phase spectrum, which contains structural information, as an explicit constraint, ensuring that the generated images exhibit stronger reliability.
- During training, our diffusion model only needs to focus on style information, effectively reducing the training burden of the model. The diffusion inference process is more efficient, allowing simultaneous training of the segmentation model and data divergence, thus reducing time overhead.

The specific workflow of this method is illustrated in Figure 8.

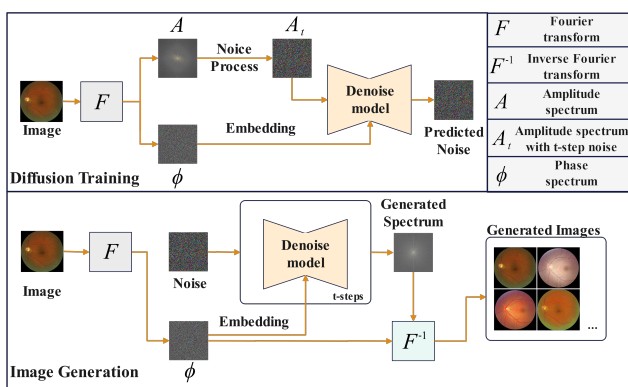

Figure 8: The workflow of the Diffusion-Based Style Divergence.

## E  STRONG AND WEAK NATURAL SELECTION STRATEGY DESCRIPTION

### E.1  WEAK NATURAL-SELECTION STRATEGY

First, during training at step $t$, we save the feature maps of the intermediate layers $l$ for both the segmentation model and the reconstruction model, denoted as $F_{seg}^{(t,l)}$ and $F_{rec}^{(t,l)}$, respectively. Then, the DNA model undergoes $k$ steps of parameter updates, yielding the corresponding intermediate layer feature maps $F_{seg}^{(t+k,l)}$ and $F_{rec}^{(t+k,l)}$. We compute the Euclidean distances in different directions after the model update, specifically $d(F_{seg}^{(t,l)}, F_{seg}^{(t+k,l)})$, $d(F_{seg}^{(t,l)}, F_{rec}^{(t+k,l)})$, $d(F_{rec}^{(t,l)}, F_{seg}^{(t+k,l)})$, and $d(F_{rec}^{(t,l)}, F_{rec}^{(t+k,l)})$. If the model maintains its evolutionary direction (e.g., $d(F_{seg}^{(t,l)}, F_{seg}^{(t+k,l)}) > d(F_{seg}^{(t,l)}, F_{rec}^{(t+k,l)})$), we define this as a recessive gene $a$; conversely, if the model shifts toward the evolutionary direction of the other model (e.g., $d(F_{seg}^{(t,l)}, F_{seg}^{(t+k,l)}) < d(F_{seg}^{(t,l)}, F_{rec}^{(t+k,l)})$), we define it as a dominant gene $A$. Based on Mendelian genetics, four combinations of gene pairs are possible, as illustrated in Figure 9.

Next, we assign different learning rates to the intermediate layers according to the proportion of dominant genes, allowing for selective local bias learning in the intermediate layers. Ultimately, we obtain a weak natural selection factor $f_{ws}^{(l)}$: when the gene combination is $AA$, $f_{ws}^{(l)} = 1$; when the combination is $Aa$, $f_{ws}^{(l)} = 1$; and when the combination is $aa$, the layer behaves as a recessive trait and does not participate in collaborative learning, with $f_{ws}^{(l)} = 0$, referred to as a "pseudogene".

### E.2  STRONG NATURAL-SELECTION STRATEGY

The strong natural-selection strategy allows DNA-Net to focus on efficiently completing the segmentation task. Meanwhile, the segmentation model can more quickly incorporate the intermediate layer mapping information from the reconstruction model during gradient descent. This approach mitigates the impact of randomness introduced by weak natural selection on the model's convergence speed, as illustrated in Figure 10.

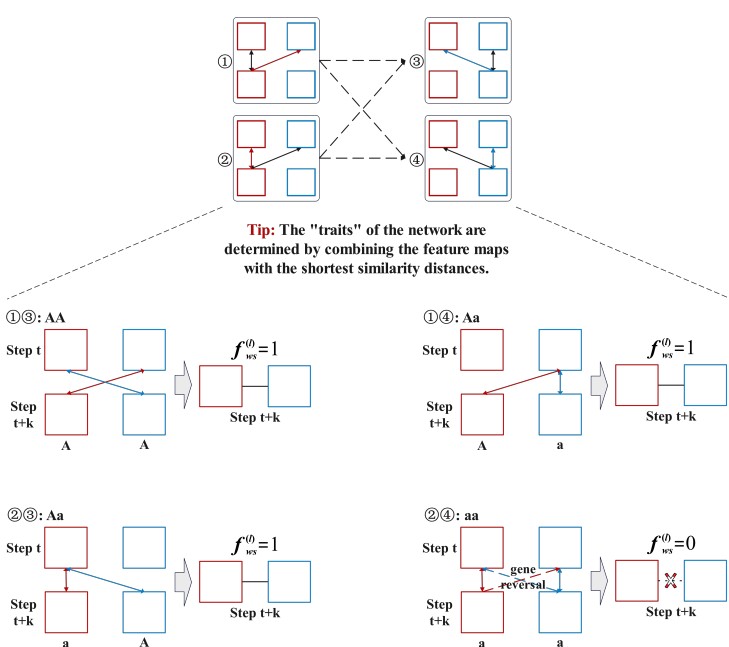

Figure 9: Examples of permutations and combinations for the weak natural selection strategy.

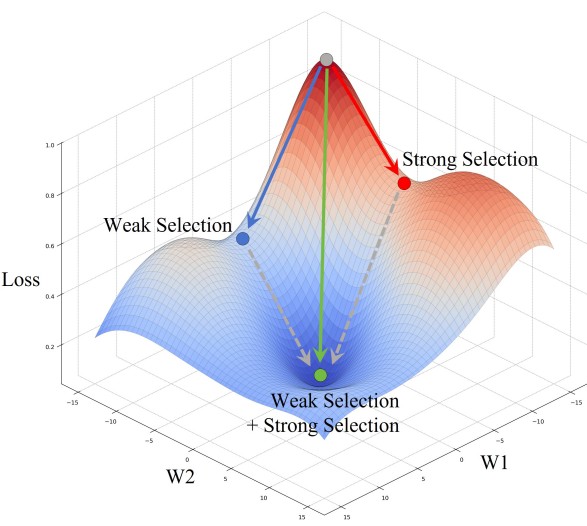

Figure 10: Illustration of loss descent. Strong natural selection enables the model to bypass the adaptiveness saddle point, allowing it to reach a high-adaptability, high-performance valley more quickly.

Specifically, we first calculate the Euclidean distances in different directions after model updates and then assess their validity based on magnitude, with smaller Euclidean distances indicating the model's evolutionary direction. The Euclidean distance corresponding to the current evolutionary direction is considered the effective Euclidean distance. We then define the sign of the Euclidean distance based on the model's direction: if the model evolves toward the reconstruction model, the effective Euclidean distance is positive; otherwise, it takes a negative value. Finally, the effective restoration Euclidean distances of the two models are summed and mapped through a Sigmoid function scaled to the range of 0-2, yielding a strong natural selection factor $f_{ss}^{(l)}$, which enables global task bias. This strategy is illustrated in Figure 11.

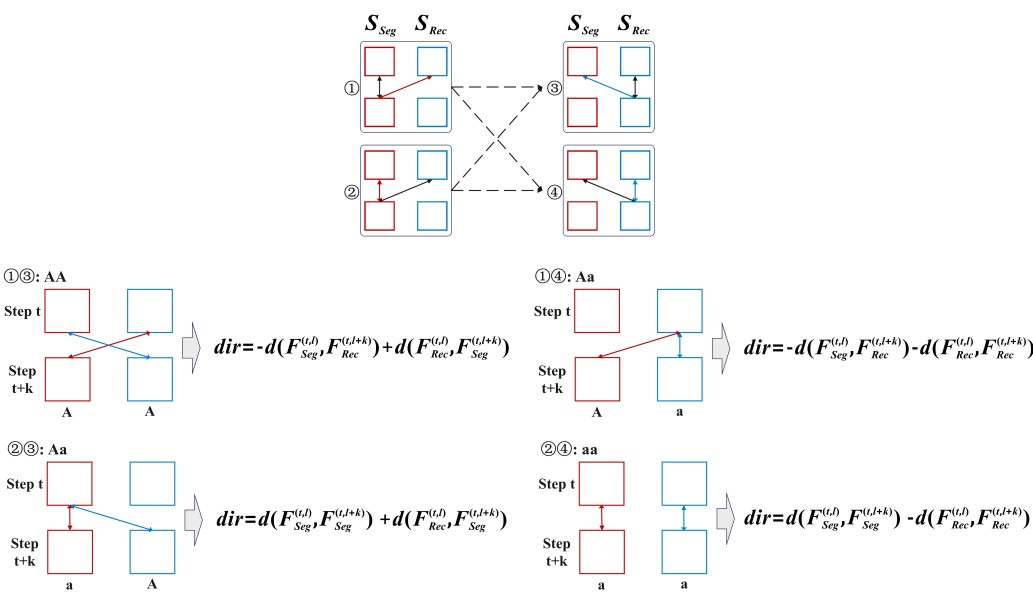

Figure 11: Examples of permutations and combinations for the strong natural selection strategy.

The algorithm descriptions of the weak natural selection and strong natural selection strategies are shown in Algorithm 1 and Algorithm 2, respectively.

---

**Algorithm 1** Weak Natural-Selection Algorithm.

---

**Input:** Training stage $t$, feature maps $F_{seg}^{(t,l)}$ and $F_{rec}^{(t,l)}$ of intermediate layer $l$; feature maps $F_{seg}^{(t+k,l)}$ and $F_{rec}^{(t+k,l)}$ after $k$-step update.

**Output:** Weak natural selection factor $f_{ws}^{(l)}$

1: $gene_{seg} \leftarrow d\left(F_{seg}^{(t,l)}, F_{seg}^{(t+k,l)}\right) > d\left(F_{seg}^{(t,l)}, F_{rec}^{(t+k,l)}\right)$

2: $gene_{rec} \leftarrow d\left(F_{rec}^{(t,l)}, F_{rec}^{(t+k,l)}\right) > d\left(F_{rec}^{(t,l)}, F_{seg}^{(t+k,l)}\right)$

$\triangleright A : gene == True. a : gene == False$

3: With probability $\delta$, $gene \leftarrow -gene$

4: $character \leftarrow gene_{seg}$ or $gene_{rec}$

5: **if** $character$ **then**

6: $\quad f_{ws}^{(l)} \leftarrow 1.0$

7: **else**

8: $\quad f_{ws}^{(l)} \leftarrow 0.0$ $\triangleright$ Referred to as a "pseudogene" Proudfoot et al. (1982)

9: **end if**

10: **return** $f_{ws}^{(l)}$

---

---

**Algorithm 2** Strong Natural-Selection Algorithm.

---

**Input:** Training stage $t$, feature maps $F_{seg}^{(t,l)}$ and $F_{rec}^{(t,l)}$ of intermediate layer $l$; feature maps $F_{seg}^{(t+k,l)}$ and $F_{rec}^{(t+k,l)}$ after $k$-step update.

**Output:** Strong natural selection factor $f_{ss}^{(l)}$

$\triangleright$ $dir1$ is the evolutionary direction of $S_{seg}$
$\triangleright$ $dir2$ is the evolutionary direction of $S_{rec}$

1: $dir_{seg} \leftarrow \begin{cases} -d\left(F_{seg}^{(t,l)}, F_{rec}^{(t+k,l)}\right) & \text{if } gene_{seg} \\ d\left(F_{seg}^{(t,l)}, F_{seg}^{(t+k,l)}\right) & \text{else} \end{cases}$

2: $dir_{rec} \leftarrow \begin{cases} d\left(F_{rec}^{(t,l)}, F_{seg}^{(t+k,l)}\right) & \text{if } gene_{rec} \\ -d\left(F_{rec}^{(t,l)}, F_{rec}^{(t+k,l)}\right) & \text{else} \end{cases}$

3: $f_{ss}^{(l)} \leftarrow \text{Sigmoid}_{0-2}(dir1 + dir2)$

4: **return** $f_{ss}^{(l)}$

---

# F    EXPERIMENTAL RESULTS

## F.1    ABLATION EXPERIMENTS OF RECONSTRUCTION LOSS

We validated the loss function of the reconstruction branch (validated on the BraTS dataset). Compared with the common L2 or MSE loss, the SSIM loss slightly improves the performance of our proposed model. However, this improvement is minimal. Therefore, considering the space limitation, we did not provide excessive introduction in the paper. The specific experimental results are shown in table 5.

Table 5: The ablation results of the reconstructed loss function.

|  | L2 | | MSE | | SSIM | |
|---|---|---|---|---|---|---|
|  | Dice↑ | HD↓ | Dice↑ | HD↓ | Dice↑ | HD↓ |
| T2 (Source) | 84.55±0.32 | 4.62±0.22 | 84.62±0.53 | 4.49±0.14 | 84.89±0.35 | 4.58±0.12 |
| T2-others (Avg) | 70.98±0.37 | 11.96±0.31 | 71.28±0.31 | 12.01±0.22 | 71.49±0.28 | 11.85±0.16 |

Among them, T2 (Source) represents the performance of the model trained by T2 in the T2 source domain, and T2 - others (Avg) represents the average value of the performance on other domains. According to our analysis, we believe that the performance improvement of the SSIM part is mainly due to its loss calculation for brightness, contrast, and structure. In such medical image reconstruction tasks with inconsistent styles, it can better capture the differences in style and perform better image reconstruction, thus bringing about a slight performance improvement. And the weighted combination of the SSIM loss and the DNA loss is to enable the reconstruction model to also learn the feature representation of the segmentation model through cooperative learning, so as to realize the joint optimization of the two. In this way, the reconstruction task is no longer to pursue the reconstruction quality independently, but to serve the segmentation model.

## F.2    COMPARATIVE EXPERIMENTAL RESULTS

We also verified the motivation of negative transfer of domain knowledge on the multi-source domain generalization dataset Fundus. From Table 6, we can see that DNA-Net can well protect the source domain knowledge on the Fundus dataset. The visualization results are shown in Figure 12.

Table 6: The source domain performance of the model on the Fundus dataset is presented below. In the Fundus dataset, Domain 1 serves as the target domain, while D.x represents Domains 2, 3, and 4.

| Fundus | | | | | | | | | | | |
|---|---|---|---|---|---|---|---|---|---|---|---|
| | | | | | | Target Domain: Domain1 | | | | | |
| Method | Dice | | ASD | | Dice | | ASD | | Dice | | ASD | |
| | Disc(D2) | Cup(D2) | Disc(D2) | Cup(D2) | Disc(D3) | Cup(D3) | Disc(D3) | Cup(D3) | Disc(D4) | Cup(D4) | Disc(D4) | Cup(D4) |
| No Generalization | 92.65 | 81.65 | 9.27 | 17.64 | 90.93 | 87.28 | 11.25 | 15.21 | 91.11 | 89.25 | 10.97 | 14.83 |
| Fed-DG Liu et al. (2021) | 86.10 | 78.12 | 15.91 | 18.46 | 90.14 | 85.17 | 11.13 | 15.97 | 84.15 | 80.32 | 17.52 | 18.10 |
| FreeSDG Li et al. (2023a) | 85.31 | 79.31 | 16.42 | 17.92 | 90.01 | 84.94 | 12.07 | 16.31 | 90.26 | 85.86 | 12.63 | 15.99 |
| SADN Zhou et al. (2022c) | 82.52 | 75.49 | 17.33 | 19.63 | 88.42 | 84.67 | 13.25 | 16.98 | 89.51 | 87.95 | 13.19 | 16.01 |
| EGSDG Jiang & Gu (2024) | 87.23 | 76.15 | 14.22 | 18.89 | 85.47 | 80.62 | 16.32 | 18.18 | 87.23 | 83.16 | 14.41 | 16.14 |
| DFQ Bi et al. (2024) | 90.57 | 77.86 | 10.93 | 18.01 | 87.72 | 83.17 | 14.20 | 15.96 | 90.57 | 84.74 | 11.36 | 17.23 |
| **DNA-Net** | **93.80** | **79.95** | **7.86** | **17.79** | **91.77** | **86.67** | **10.55** | **15.52** | **92.44** | **88.45** | **10.76** | **15.57** |

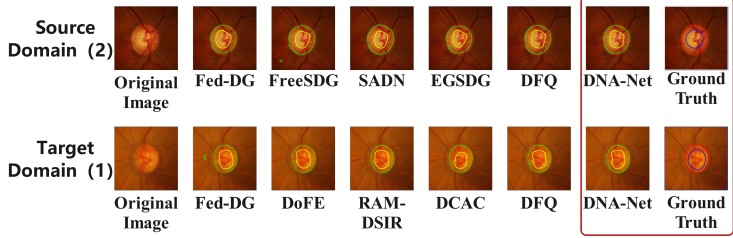

Figure 12: Visual comparison of model prediction results on Domain 2 source domain and Domain 1 target domain.