# OpenReview forum: "DNA-Net: Genetic-Inspired Dual-Chain Learning for Medical Image Domain Generalization without Negative Transfer"
_ICLR.cc/2026/Conference — ICLR 2026 Conference Withdrawn Submission_

### Official Review · Reviewer_n7RD · 2025-10-29

**Soundness:** 3
**Presentation:** 2
**Contribution:** 2
**Rating:** 4
**Confidence:** 5

**Summary:**

This paper proposes a new solution for domain generalization in medical image segmentation, including three critical designs:
(1) A structure-guided augmentation using Bezier curves.
(2) A conditional diffusion-based style divergence model that performs perturbations on the amplitude spectrum to enhance style diversity while preserving anatomical structures constrained by the phase spectrum.
(3) A genetic-inspired DNA-Net that employs weak and strong natural selection mechanisms to guide collaborative learning between segmentation and reconstruction tasks.
Experiments on two benchmarks, including BraTS and optic disc/cup segmentation tasks, show that the proposed method achieves superior cross-domain performance without negative transfer on the source domain.

**Strengths:**

1. This paper introduces a diffusion-based style augmentation that explicitly separates the amplitude and phase components of the Fourier spectrum and performs perturbations on the amplitude spectrum to enhance style diversity.
2. The proposed DNA-Net incorporates biological concepts such as weak and strong natural selection into a dual-chain collaborative learning framework.

**Weaknesses:**

1. The implementation details are insufficiently introduced. The specific network architecture of the DNA-Net and the particular layers of features used to compute the similarity remain unclear.
2. The values of the hyperparameters seem to be carefully adjusted. However, there is no experiment discussing the hyperparameter sensitivity.
3. Lack of comparison with the state-of-the-art methods.
4. The proposed framework is much complex, but the performance gains are not significant, particularly on the optic disc/cup segmentation task. It also lacks a complexity analysis.
5. In Table 1, several competing methods achieve worse performance than the baseline, and there is no explanation for this phenomenon.
6. In Figure 3, the subscript of the sign representing the blue box should be "Rec".
7. It lacks the reproducibility statement for the compared methods.

**Questions:**

1. Please elaborate on the implementation details of the proposed framework.
2. Please conduct a discussion on the selection of hyperparameters.
3. Please compare the proposed method with the latest works published in 2025.
4. Please conduct a complexity analysis on the proposed method compared with others.

---

### Official Review · Reviewer_Q53n · 2025-10-29

**Soundness:** 2
**Presentation:** 2
**Contribution:** 2
**Rating:** 4
**Confidence:** 5

**Summary:**

This paper proposes DNA-Net, a genetic-inspired dual-chain collaborative learning framework, to address the challenges of limited target domain coverage and inter-domain negative transfer in medical image domain generalization (DG). The work demonstrates notable innovations and solid experimental support, though several aspects merit further refinement to enhance its rigor and impact.

**Strengths:**

The concept of modeling the learning framework after DNA's dual-chain structure is highly creative and represents a fresh perspective in the domain generalization literature.
The paper directly tackles the often-overlooked issue of negative transfer in domain generalization, where improving performance on unseen target domains comes at the cost of degrading performance on the source domain. This is a crucial consideration for real-world clinical deployment.
The combination of frequency-domain diffusion for style augmentation and the dual-chain collaborative learning with a natural selection-inspired loss function is technically sound and appears effective.

**Weaknesses:**

The Bezier curve-based pixel-level nonlinear transformation in the Broad Domain Style Exploration module is introduced, but the paper only provides a formula (without defining key parameters like the number of control points pi and their setting criteria) and lacks analysis of how different curve degrees or control point configurations affect style exploration breadth. Supplementary experimental comparisons or parameter sensitivity analyses would strengthen the module's rationality.
In the weak natural selection strategy, the "gene mutation probability \delta
" is set to 0.05, but the paper does not explain why this value is chosen or how \delta influences model stability (e.g., whether overly high
\delta  disrupts collaborative learning or overly low \delta limits trait diversity). Validation of multiple \delta values would improve the strategy’s robustness.

**Questions:**

The paper is generally well-written. A thorough proofreading is recommended to catch minor typos (e.g., "divergence" in Figure 8 caption).
Consider moving the highly relevant "Use of LLMs" section from the appendix to the main reproducibility statement (Section 7).
While the paper mentions that amplitude spectrum diffusion reduces computational cost compared to direct image diffusion, it lacks quantitative metrics (e.g., FLOPs, inference time, GPU memory usage) for the full DNA-Net model and comparisons with SOTA methods.

---

### Official Review · Reviewer_a5WY · 2025-10-30

**Soundness:** 2
**Presentation:** 2
**Contribution:** 1
**Rating:** 2
**Confidence:** 4

**Summary:**

This paper proposes a novel framework, DNA-Net, to address the critical challenge of Domain Generalization (DG) in medical image segmentation, with a specific focus on mitigating negative transfer (where improving performance on unseen target domains degrades performance on the source domain). Extensive experiments on BraTS (brain MRI) and a fundus image dataset demonstrate that DNA-Net achieves state-of-the-art performance on both unseen target domains and, crucially, maintains or even improves performance on the source domain, effectively suppressing negative transfer.

**Strengths:**

+ This paper explicitly addresses the often-overlooked issue of "negative transfer" in DG, which is a significant practical concern for clinical deployment. Protecting source domain performance is crucial for maintaining trust and safety when a model is applied to new hospitals.
+ The paper provides strong empirical evidence through: Comparisons with multiple SOTA methods on two different medical imaging benchmarks (brain MRI and fundus).Evaluation on both single-source and multi-source DG settings. Thorough ablation studies that validate the contribution of each proposed component (Style Exploration, Style Divergence, Weak/Strong Selection).

**Weaknesses:**

+ The description of the Weak Natural-Selection Strategy, particularly the definitions of "dominant gene A" and "recessive gene a," is somewhat confusing and hard to follow in the main text. While the appendix provides algorithms, the core intuition could be better explained in the main body.
+ While the genetic analogy is inspiring, the paper could do more to justify why this specific biological model is particularly suited to the DG problem, beyond the high-level concept of collaboration and selection. A deeper discussion linking the biological principles to the technical mechanics would strengthen the narrative.
+ The ablation study (Table 4) convincingly shows the contribution of each module. However, it would be even stronger if it included a baseline that uses a simpler, non-genetic collaborative learning approach (e.g., a simple shared encoder or a standard multi-task learning setup) to more directly isolate the benefit of the novel natural selection strategies.
+ There are a few minor grammatical errors and awkward phrasings (e.g., "The trained models have a large number of features, but it can be easily seen that the trained models perform better..." on Page 8 is redundant).
+ Some references in the main text are missing from the bibliography (e.g., Ma et al. (); Li et al. (); on Page 2, Zhou et al. () on Page 4). This should be corrected.

**Questions:**

See above

---

### Official Review · Reviewer_VCSh · 2025-11-01

**Soundness:** 2
**Presentation:** 2
**Contribution:** 2
**Rating:** 2
**Confidence:** 4

**Summary:**

Authors present a novel augmentation strategy for medical image
analysis. Specifically, they first use a traditional approach to
augment a given dataset. Samples in the augmented dataset is divided
to amplitude and phase signals using the Fourier transform. The
amplitude is then varied using a diffusion model. The variations are
used with the phase images to generate new samples, effectively
extending the dataset even further.
In addition, the authors propose a dual branch network for tackling
two tasks together: source domain image restoration and generalization
segmentation (for which an explicit definition is not given in the
article). Through this, authors claim that they tackle inter-domain
negative transfer during generalization, which is a new term authors
introduce but I think it is the same as "forgetting" of continual
learning. Experiments are presented using the BraTS and fundus imaging
datasets.

**Strengths:**

- Domain generalization is an important problem in medical image
  analysis.
- Using reconstruction, defined as going from augmented samples to the
  original, as a side task for segmentation is an interesting and
  novel, to the best of my knowledge, approach.
- Using the phase as a condition for a diffusion model to generate new
  amplitude images is an interesting and, to the best of my knowledge,
  novel approach.

**Weaknesses:**

- The level of novelty is rather weak.
  + Augmentation for domain generalization has been well studied.
  + Augmentation using Fourier transform has been well studied.
  + Multi-task architectures utilizing multi-branch networks have been
    well studied.
- What author defines as "inter-domain negative transfer" is well
  studied in the continual learning. This problem is often studied
  separate than domain generalization. I strongly recommend authors to
  check that field for very relevant works. They do not need to define
  a new term, they can use the term "forgetting" or "catastrophic
  forgetting". *It is important to position the work within that
  literature.*
- The claim that $D^{S'}$ incorporates source and target domain
  distributions is not a substantiated one. Authors simply use
  augmentation. The mentioned incorporate is a hope not a given fact.
- Dual branch networks have been used quite a lot in multi-task
  learning and other problems. I believe the "DNA-net" is such an
  architecture. *It is important to position the work within that
  literature.* As far as I can see, authors are using a dual branch
  architecture with feature-level integration between the two
  branches.
- Description of the DNA-Net is full of analogies. This makes it very
  difficult for readers to understand. It is extremely important that
  authors provide a mathematical description of the model. The
  equations given in Figure 3 are naturally not enough to describe the
  strategy in detail.
- The domain generalization setup in the BraTS dataset is very
  superficial. Please use the SSA part of the BraTS dataset, which
  actually provides a realistic domain shift. Furthermore, it seems
  like the experiments with the BraTS dataset are done in 2D with a
  single slice. This reduces the realism of the experiments.

**Questions:**

- Authors state that the proposed frequency-domain diffusion approach
  mitigates the high computational cost and memory demands associated
  with performing direct diffusion generation on medical
  images. However, the amplitude of the Fourier transform of an image
  is of the same size as the image. How does this lead to any
  reduction in computational costs ?
- There are two claimed contributions in this article. The first is
  the augmentation and the second is the DNA-Net. In this setup, I
  think it makes sense to run the other methods with the same
  augmentation as DNA-Net so that the isolated contribution of DNA-Net
  can be quantified.

---

### Note · Authors · 2025-11-13

I have read and agree with the venue's withdrawal policy on behalf of myself and my co-authors.